# Fluorescein-based sensors to purify human α-cells for functional and transcriptomic analyses

Sevim Kahraman[1,2,3], Kimitaka Shibue[1,2,3], Dario F De Jesus[1,2,3], Hyunki Kim[1,2,3], Jiang Hu[1], Debasish Manna[4,5], Bridget Wagner[4], Amit Choudhary[4,5], Rohit N Kulkarni[1,2,3]*

[1]Islet Cell and Regenerative Biology, Joslin Diabetes Center, Boston, United States; [2]Department of Medicine, Beth Israel Deaconess Medical Center, Harvard Medical School, Boston, United States; [3]Harvard Stem Cell Institute, Harvard Medical School, Boston, United States; [4]Chemical Biology and Therapeutics Science Program, Broad Institute of MIT and Harvard, Cambridge, United States; [5]Divisions of Renal Medicine and Engineering, Brigham and Women's Hospital, Boston, United States

**Abstract** Pancreatic α-cells secrete glucagon, an insulin counter-regulatory peptide hormone critical for the maintenance of glucose homeostasis. Investigation of the function of human α-cells remains a challenge due to the lack of cost-effective purification methods to isolate high-quality α-cells from islets. Here, we use the reaction-based probe diacetylated Zinpyr1 (DA-ZP1) to introduce a novel and simple method for enriching live α-cells from dissociated human islet cells with ~95% purity. The α-cells, confirmed by sorting and immunostaining for glucagon, were cultured up to 10 days to form α-pseudoislets. The α-pseudoislets could be maintained in culture without significant loss of viability, and responded to glucose challenge by secreting appropriate levels of glucagon. RNA-sequencing analyses (RNA-seq) revealed that expression levels of key α-cell identity genes were sustained in culture while some of the genes such as *DLK1*, *GSN*, *SMIM24* were altered in α-pseudoislets in a time-dependent manner. In conclusion, we report a method to sort human primary α-cells with high purity that can be used for downstream analyses such as functional and transcriptional studies.

*For correspondence: rohit.kulkarni@joslin.harvard.edu

## Editor's evaluation

The manuscript by Kahraman et al. describes the use of the fluorogenic dye – diacetylated Zinpyr1 (DA-ZP1) – for purifying cadaveric human islet α cells. The data show that DA-ZP1 is a useful non-antibody-based approach to label α cells and provides additional evidence that the purified α cells remained viable and functional after several days in culture. This resource will be a useful tool for islet biologists and researchers investigating islet cell dysfunction in diabetes.

## Introduction

Zinc-binding molecules such as Newport Green (NPG) (*Kirkpatrick et al., 2010*) and ZIGIR (zinc granule indicator) (*Ghazvini Zadeh et al., 2020*) have been employed previously for the purification of human or murine α-cells and β-cells. While NPG was reported to sort cell population with 75% enriched α-cells, ZIGIR enabled isolation of >95% pure α-cells when used in combination with anti-bodies specific to endocrine cells and α-cells such as HPi2 (*Dorrell et al., 2008*) and TM4S4F (*Muraro et al., 2016*), respectively. We have recently reported the purification of human pancreatic β-cells (*Lee*

et al., 2020) and stem-cell derived β-like cells (Kahraman et al., 2021) using the zinc-based reaction probe diacetylated Zinpyr1 (DA-ZP1). DA-ZP1 is a non-fluorescent zinc sensor that binds Zn(II) with nanomolar affinity (Chyan et al., 2014). Binding of Zn(II) selectively and rapidly mediates hydrolytic cleavage of the acetyl groups and generates a strong fluorescence to sort the labeled cells by fluorescence activated cell sorting (FACS). In Lee et al., we performed FACS analysis of DA-ZP1-stained human islets cells and observed enrichment of β-cells in the DA-ZP1 positive population (Lee et al., 2020). To enhance purity of sorted human β-cells, we used a conservative gating strategy and sorted the cell population that emitted high DA-ZP1 fluorescence while excluding the other cell populations with low or no DA-ZP1 fluorescence. In the present study, we modified the gating strategy to explore whether zinc probes can be used to identify a purified population of α-cells within a mixed population of pancreatic endocrine cells. We reclassified DA-ZP1 positive cells as either 'DA-ZP1 intermediate' or 'DA-ZP1 bright' by distinctly marking the boundary among the fractions to contrast those with 'bright' DA-ZP1 fluorescence that were identified as β-cells. These data indicate that high purity α-cells and β-cells can be generated by simultaneously sorting islet cells using a zinc reaction probe (DA-ZP1). This approach could be useful for studying purified primary human α-cells for functional analyses and for comprehensive evaluation of the transcriptomes to further increase our understanding of α-cell biology in health and disease.

## Results

### DA-ZP1 as a tool to purify live human pancreatic α-cells

To test whether DA-ZP1 is able to sort α-cells from a mixed population of pancreatic endocrine cells, we dispersed human islets into single cells and labeled them with DA-ZP1 (Figure 1a). Flow cytometry analysis of the cells showed a wide spread of fluorescence intensity among dispersed islet cells on a two-dimensional density plot. We classified the cells into three subsets based on their fluorescence intensity and drew a gate to separate each of the subsets (Figure 1b). The subset centered near the unstained cell background showed 'low' fluorescence intensity while the other two cell populations positioned on the right side of the dot-plot showed 'intermediate' and 'bright' fluorescence intensities, respectively (Figure 1a and b). We confirmed that DA-ZP1 labeling resulted in a similar fluorescence intensity pattern among the three cell populations (low, intermediate, bright) in an additional four independent human islet donors (Figure 1—figure supplement 1). Consistent with the FACS-based fluorescence assessment, fluorescent microscopy validated that the DA-ZP1 'bright' subset displayed higher fluorescence signal compared to the intermediate subset, and that the unsorted islet cells are comprised of a mixture of cells with varying fluorescence intensities (Figure 1c, Figure 1—figure supplement 1). To identify percentage of hormone-containing cells in each subset, the sorted cells were plated, fixed, and immunostained using antibodies to detect insulin or glucagon. Immunofluorescence analysis showed that the bright subset was highly enriched for human β-cells (~83% CPEP+ cells), while the intermediate subset consisted of α-cells of high purity (~95% GCG+ cells) (Figure 1d and e). The cells in the DA-ZP1 'low' subset were mostly hormone negative, indicating that these cells likely represented non-hormonal cells such as fibroblast-like, endothelial, or exocrine cells. Consistently, transcriptomics analysis of three different populations by RNA-seq showed that the expression levels of α-cell markers such as GCG, TTR, ARX, IRX1, IRX2 are higher in the intermediate subset, the expression levels of β-cell markers such as INS, MAFA, PDX1, IAPP are higher in the bright subset, and the expression levels of non-endocrine cell markers such as CFTR, KRT19, VIM are higher in the low subset compared to other subsets (Figure 1f). The low expression levels of SST, HHEX, GHRL, and PPY in the intermediate subset compared to other subsets indicate a small amount of contamination with the other endocrine islet cell types such as delta, epsilon, and PP cells.

Next, we compared DA-ZP1 with other approaches that utilize zinc-based molecules for their ability to sort human α-cells. While DA-ZP1 generated three distinct clusters of cells with low, intermediate, or bright fluorescence, ZIGIR-stained cells and NPG-stained cells formed a single bright cluster with no distinct intermediate population to separate α-cells from β-cells easily (Figure 1g). These data show that DA-ZP1 is unique in its ability to label α-cells without the aid of an antibody-based approach making it superior to other zinc-based molecules.

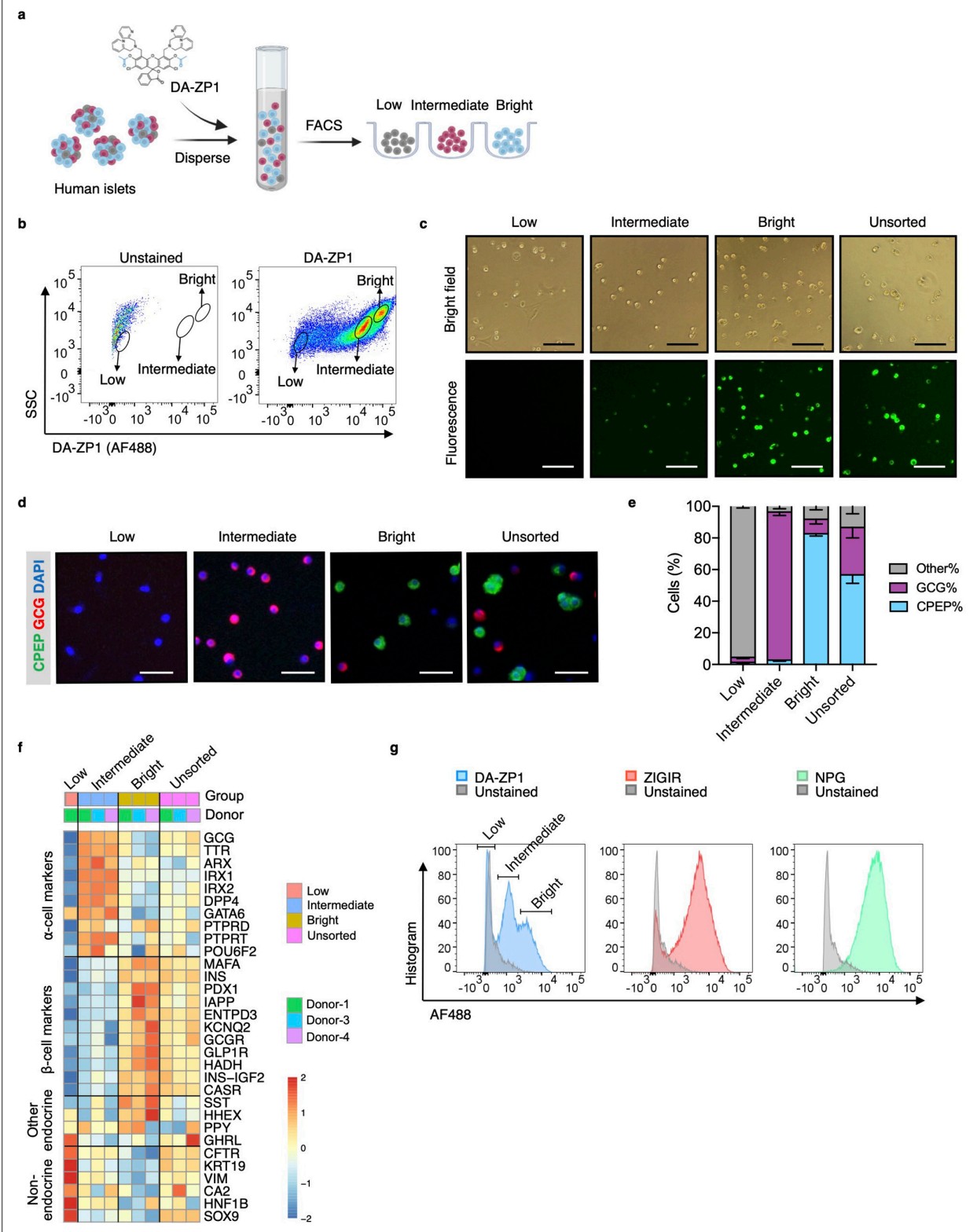

**Figure 1.** Isolation of live human pancreatic α-cells after staining with diacetylated Zinpyr1 (DA-ZP1) by fluorescence activated cell sorting (FACS). (**a**) Experimental outline. (**b**) Representative FACS plot showing three cell populations with low, intermediate, or bright fluorescence. The plot represents the data collected from Donor-1 islets. Unstained (left) vs DA-ZP1-treated (right) human islets. Gating strategy and the data collected from the other donors (n=4) are given in ***Figure 1—figure supplement 1***. (**c**) The DA-ZP1 derived green fluorescence is maintained in the next day of sorting in the sorted islet cells. The cells were plated in Matrigel-coated flat-bottom plates. Scale bar, 100 µm. See also ***Figure 1—figure supplement 1***. (**d**)

*Figure 1 continued on next page*

Figure 1 continued

Representative images of human islet cells after FACS showing C-peptide (green) and glucagon (red) expressing islet cells. Nuclei were stained with DAPI (blue). Scale bar, 50 µm. (**e**) Quantification of percentage of CPEP+, GCG+, and other cells (CPEP- GCG-) in each cell population. Data are presented as mean values ± s.e.m. n=3 donors. (**f**) Heatmap showing expression of genes in different cell subsets. n=3 donors. (**g**) Comparison of other zinc-based dyes with DA-ZP1 by FACS.

The online version of this article includes the following source data and figure supplement(s) for figure 1:

**Figure supplement 1.** Gating strategy for isolation of live human pancreatic α-cells.

**Figure supplement 1—source data 1.** Number of live cells collected by FACS using 15,000 islet equivalents (IEQs) determined by trypan blue staining.

## α-Pseudoislets can be maintained in culture without losing their viability

Purified pancreatic α-cells were maintained in culture for the assessment of viability post-sorting (***Figure 2a***). Both the sorted α-cells and the unsorted islet cells formed islet-like clusters shortly after seeding in round-bottom non-treated plates (***Figure 2b and c***). We refer to sorted α-cells (DA-ZP1 intermediate cells) as α-pseudoislets on day 5 and 10 post-sorting, since they consisted of highly purified GCG+ cells and formed islet-like cell clusters. Similarly, the unsorted islet cells are referred to as unsorted pseudoislets on day 5 and 10 post-sorting since they formed islet-like cell clusters. The size of the clusters was proportional to the number of cells in each well. Interestingly, α-pseudoislets tended to form tighter clusters shortly after plating compared to unsorted pseudoislets (***Figure 2—figure supplement 1***). Cell viability was determined by measuring intracellular ATP levels on day 5 and day 10 post-sorting. Culturing cells for up to 10 days did not alter viability of α-pseudoislets and unsorted pseudoislets indicating that these cells can be maintained in culture without significant cell loss. In contrast, native islets started to die after day 5 possibly due to necrosis in the core of the islets caused by hypoxia (***Giuliani et al., 2005***; ***Komatsu et al., 2017***; ***Figure 2d***). We further validated that α-pseudoislets consisted of a highly pure population of α-cells with ~95% GCG+ cells on days 5 and 10 (***Figure 2e and f***). To assess α-cell death, apoptotic index was measured by quantification of the percentage of TUNEL+GCG+ cells. Apoptosis index remained stable over the duration of the culture in both α-pseudoislets and in unsorted pseudoislets indicating that α-cells survived even after they form pseudoislets (***Figure 2g and h***). However, the percentage of apoptotic α-cells in native islets tend to increase on day 10 compared to day 5 which could be due to hypoxia (***Komatsu et al., 2017***). Notably, we observed proliferating α-cells in α-pseudoislets, unsorted pseudoislets, and native islets on both days 5 and 10 (***Figure 2i***). Percentage of Ki67+GCG+ cells was similar in each group on day 5 (α-pseudoislets; 0.038%±0.008, unsorted pseudoislets; 0.049%±0.007, native islets; 0.031%±0.004) and did not alter significantly with time spent in culture indicating that α-cells maintained their proliferation potential in culture (***Figure 2j***). The relevance of this interesting observation requires further investigation.

## α-Pseudoislets showed glucose-responsive glucagon release

Next, to investigate functional integrity, α-pseudoislets, unsorted pseudoislets, and native islets were each independently challenged with either low (3.3 mM) or high (16.7 mM) glucose after preincubation in 16.7 mM glucose (***Figure 3a***). Glucagon release significantly decreased in response to high glucose treatment in each group (***Figure 3b***), while the fold increase between low (3.3 mM) or high (16.7 mM) glucose were comparable among the groups (***Figure 3c***). Unsorted pseudoislets displayed higher basal secretion of glucagon compared to native islets which is consistent with the previous observation (***Reissaus and Piston, 2017***). α-Pseudoislets showed higher glucagon content compared to unsorted pseudoislets (***Figure 3d***), indicating high purity of sorted glucagon positive cells. These results indicate that α-pseudoislets are similar to unsorted pseudoislets in their capacity for glucose-responsive glucagon secretion.

## α-Pseudoislets maintain α-cell identity in culture

To explore whether α-pseudoislets maintain α-cell identity by preserving genes enriched in α-cells, we compared the RNA-sequencing (RNA-seq) data from α-pseudoislets, unsorted pseudoislets, and native islets on days 0, 5, 10. We first used a publicly available single-cell RNA-seq (scRNA-seq) database performed on cadaveric human islets (GSE84133) (***Baron et al., 2016***) to identify genes that are

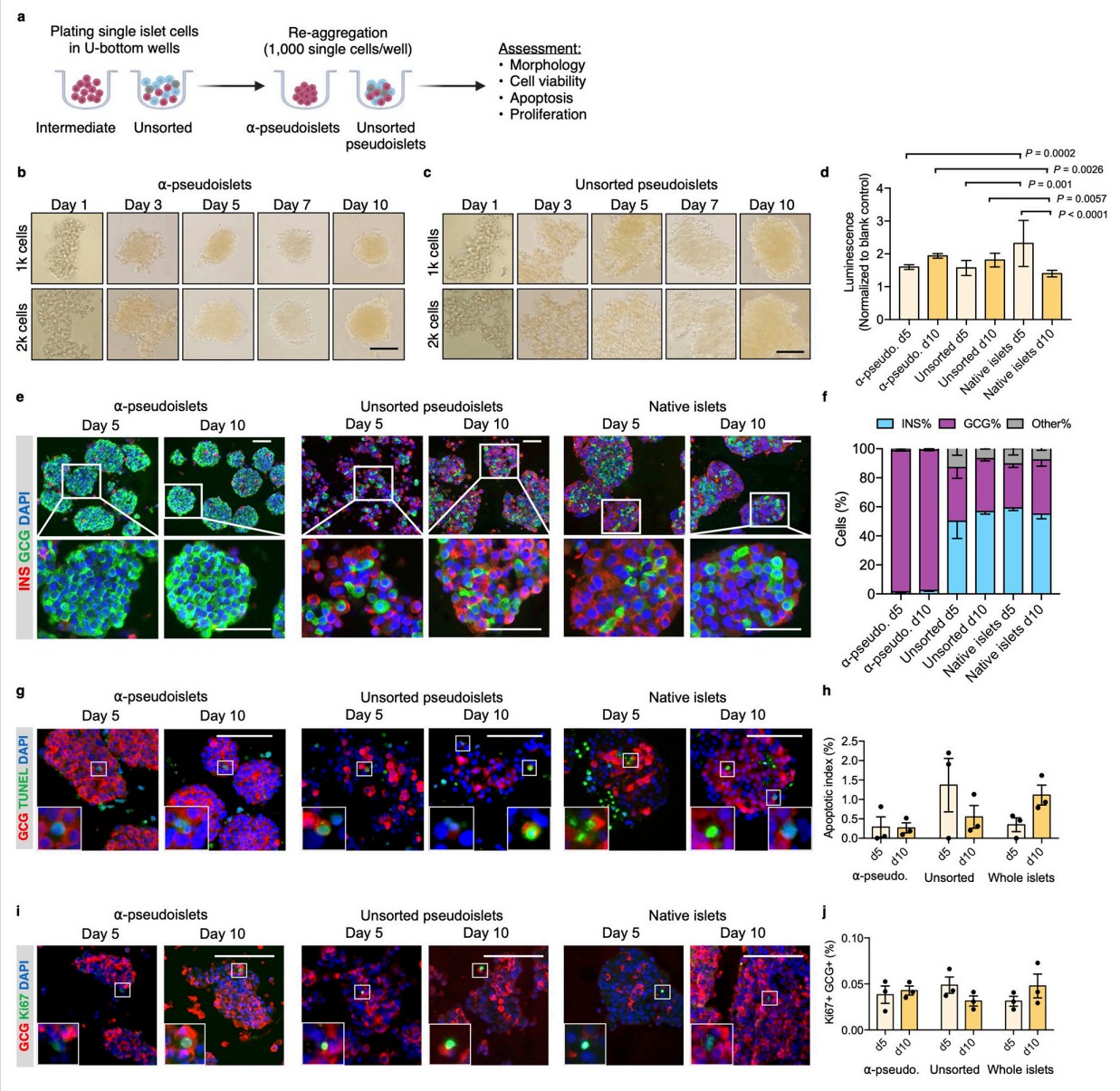

**Figure 2.** α-Pseudoislets are viable and able to proliferate in vitro post-sorting. (**a**) The single islet cells were seeded in round-bottom wells (1k cells per well) after sorting to allow re-aggregation. (**b, c**) Bright-field images of intermediate (sorted α-cells) (**b**) and unsorted pseudoislets (**c**) post-sorting. 1k (top panel) or 2k (bottom panel) single cells were seeded per well. Scale bar, 100 μm. See also *Figure 2—figure supplement 1*. (**d**) Cell viability was quantified by luminescence reflecting intracellular ATP levels on days 5 and 10 following fluorescence activated cell sorting (FACS). Fold-change relative to blank control. n=7–9 replicates using islet cells from two donors. (**e**) Representative immunostaining images of α-pseudoislets, unsorted pseudoislets, and native islets on days 5 and day 10 showing INS (red), GCG (green). Nuclei stained with DAPI are blue. For top and bottom images, scale bar, 100 μm. (**f**) Percentage of INS+, GCG+, and other (INS-GCG-) islet cells. n=3 donors. (**g**) Representative immunostaining images of α-pseudoislets, unsorted pseudoislets, and native islets on day 5 and day 10 showing GCG (red), TUNEL (green). Nuclei stained with DAPI are blue. Scale bar, 100 μm. Boxes show apoptotic α-cells. (**h**) Percentage of TUNEL+GCG+ cells. n=3 donors. (**i**) Representative immunostaining images of α-pseudoislets, unsorted pseudoislets, and native islets on day 5 and day 10 showing GCG (red), Ki67 (green). Nuclei stained with DAPI are blue. Scale bar, 50 μm. Boxes show proliferating α-cells. (**j**) Percentage of Ki67+GCG+ cells. n=3 donors. Data are presented as mean values ± s.e.m (**b–j**). n=3 donors. Two-way ANOVA followed by Sidak's multiple comparison test (**d, h, j**).

The online version of this article includes the following figure supplement(s) for figure 2:

**Figure supplement 1.** α-Pseudoislets tended to form tighter clusters.

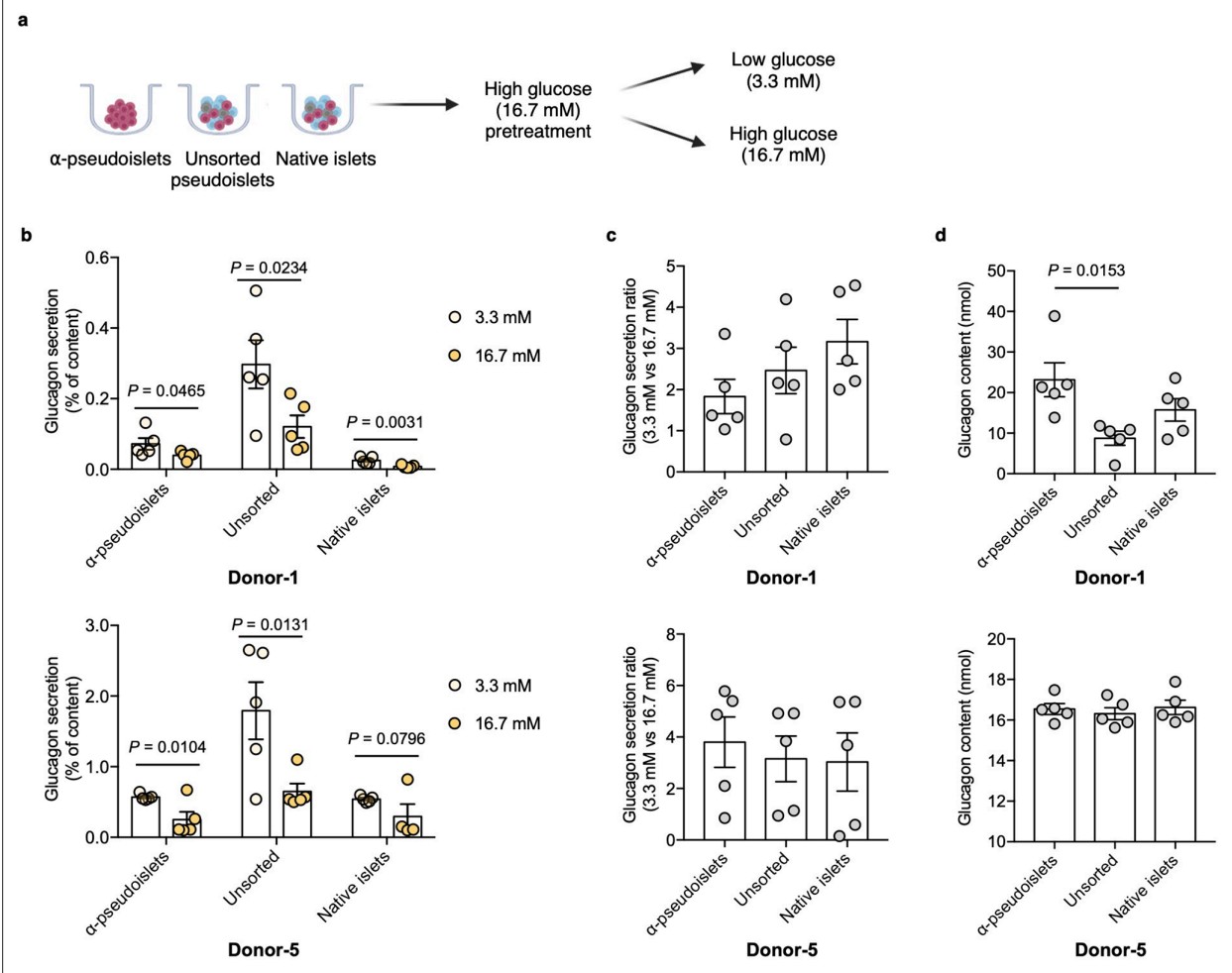

**Figure 3.** Glucagon secretion in response to glucose challenge. (**a**) α-Pseudoislets, unsorted pseudoislets, or native islets were preincubated in Krebs-Ringer bicarbonate (KRB) buffer with 16.7 mM glucose followed by the incubation in KRB buffer with 3.3 mM glucose and 16.7 mM glucose on day 5 post-sorting. (**b**) Glucagon secretion in response to glucose challenge (3.3 mM vs 16.7 mM). One-tailed Student's t-test. (**c**) Ratio of glucagon released by each groups of cells at 16.7 mM glucose versus that at 3.3 mM glucose. (**d**) Glucagon content measured in each well containing ~8000 cells (eight α-pseudoislets, eight unsorted pseudoislets, and eight native islets). Data are presented as mean values ± s.e.m. (**b–d**). n=5 replicates using islet cells from two donors (**b–d**). One-way ANOVA corrected for Tukey applied to (**c, d**).

differentially expressed between cadaveric islet α-cells and β-cells (α-cell enriched and β-cell enriched genes). Analysis of scRNA-seq data revealed 75 α-cell enriched genes and 68 β-cell enriched genes in cadaveric islet α-cells and β-cells, respectively (false discovery rate [FDR]<0.1, fold change [FC]>1.5, *Supplementary file 1*). Expression analysis of α-cell enriched and β-cell enriched genes in α-pseudoislets showed that 20 (27%) of the 75 α-cell enriched genes including *GCG, TTR, IRX2* were upregulated and 24 (35%) of the 68 β-cell enriched genes including *INS, IAPP, MAFA, PDX1, NKX6-1, G6PC2* were downregulated in α-pseudoislets compared to unsorted pseudoislets or native islets on day 0 (FDR < 0.1; FC > 2 or FC < –2, *Supplementary file 2*), which confirms the enrichment of α-cells in α-pseudoislets (*Figure 4a, Figure 4—figure supplement 1*). Concurrently, 61 (~81%) of the 75 α-cell enriched genes did not alter in α-pseudoislets on day 5 and day 10 compared to day 0 (FDR < 0.1; FC < –2 and FC > 2, *Supplementary file 3, Figure 4b*), indicating the majority of the α-cell enriched genes including *GCG, ARX, IRX2,* and *TTR* were preserved in culture. Similarly, 53 (71%) of the 75 α-cell enriched genes and 49 (72%) of the 68 β-cell enriched genes did not alter in unsorted pseudoislets on day 5 compared to day 0. However, these genes differentially expressed in unsorted pseudoislets on day 5 compared to day 0 were unchanged on day 10 compared to day 0. This might indicate that existence of other islet cells and paracrine interactions in the unsorted pseudoislets were important for maintenance of α-cell or β-cell identity (*Cigliola et al., 2018; Figure 4b, Supplementary file 3*). On

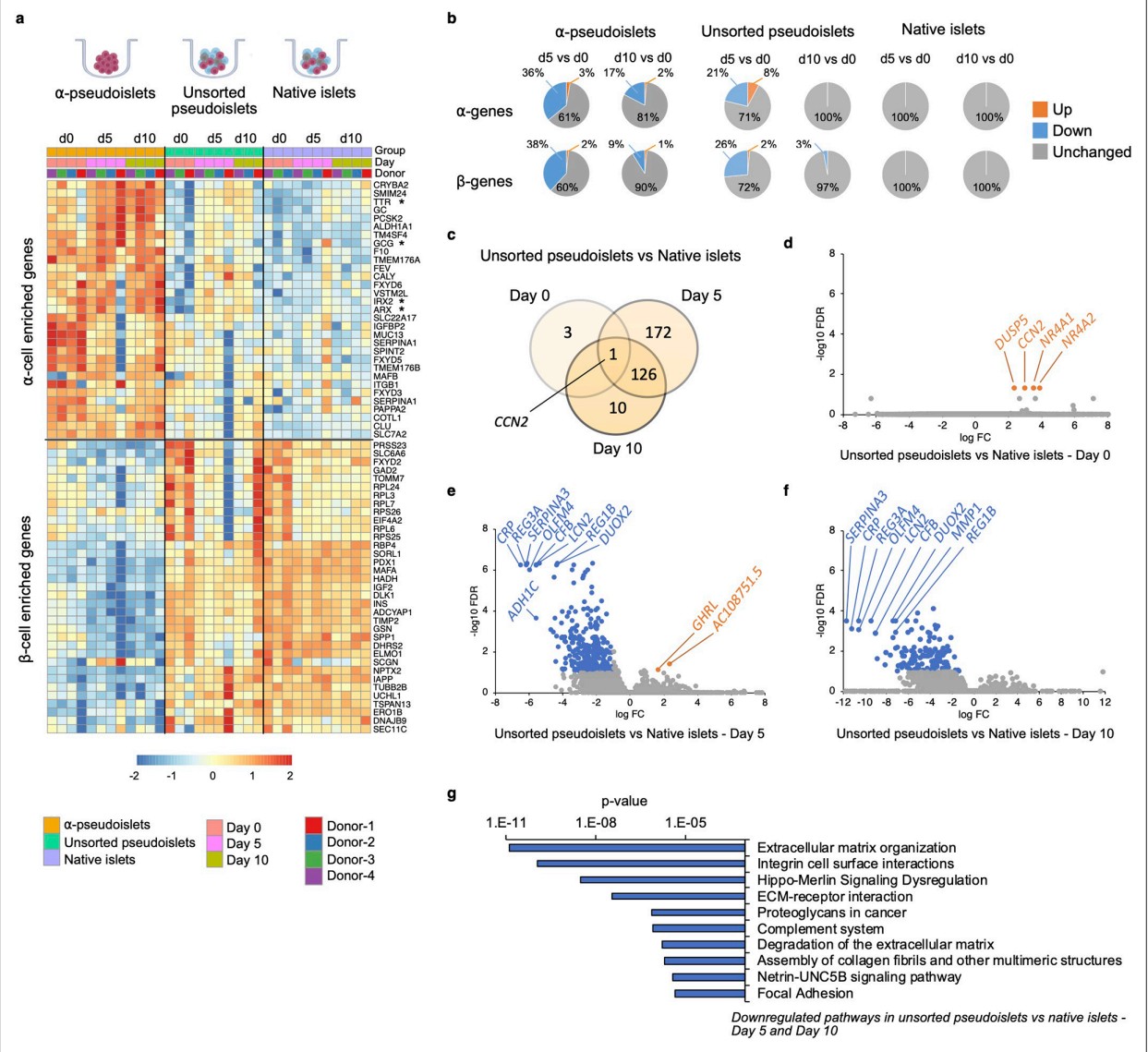

**Figure 4.** Changes in gene expression levels driven by dissociation and re-aggregation of human islet cells. (**a**) Heatmap showing expression levels of α-cell enriched and β-cell enriched genes in α-pseudoislets, unsorted pseudoislets, and native islets on days 0, 5, and 10. Asterisks show genes associated with α-cell identity and function (GCG, ARX, IRX2, TTR). (**b**) Pie charts showing percentage of α-cell enriched (top panel) and β-cell enriched (bottom panel) genes that alter in α-pseudoislets, unsorted pseudoislets, and native islets on day 5 or day 10 compared to day 0. See also *Figure 4—figure supplement 1*. (**c**) Transcriptome of unsorted pseudoislets was compared with native islets on days 0, 5, 10. Venn diagram shows number of differentially expressed genes (DEGs) between unsorted pseudoislets and native islets on different days. (**d–f**) Volcano plots showing genes downregulated (blue) or upregulated (orange) significantly (FC < –2 or FC > 2, respectively, FDR < 0.1) on day 0 (**d**), day 5 (**e**), and day 10 (**f**). Gray shows non-significant genes with FDR > 0.1 and –2 < FC < 2. (**g**) Top 10 pathways downregulated in unsorted pseudoislets on day 5 and day 10 compared to native islets. n=4 donors; α-pseudoislets d0, d5, d10, unsorted pseudoislets d5, native islets d5, d10, and n=3 donors; unsorted pseudoislets d0, d10, native islets d0 (**a–g**).

The online version of this article includes the following figure supplement(s) for figure 4:

**Figure supplement 1.** Changes in expression levels of α-cell enriched and β-cell enriched genes in α-pseudoislets, unsorted pseudoislets, and native islets on days 0, 5, 10.

the other hand, native islets did not show any changes in expression levels of α-cell enriched or β-cell enriched genes on days 5 and 10 compared to day 0 indicated that maintenance of islet structure was likely necessary to maintain cell identity. In sum, comparison of α-pseudoislets, unsorted pseudoislets, and native islets on days 0, 5, 10 showed that co-existence of other cells and intact islet architecture were desirable but not indispensable for maintenance of α-cell identity in vitro.

## Extracellular matrix organization genes are downregulated in dissociated and re-aggregated islet cells

To investigate the potential transcriptional changes that occur secondary to cell-to-cell interactions, we compared the transcriptome of native islets with that of unsorted pseudoislets on days 0, 5, and 10 of culture (*Figure 4c*). We filtered differentially expressed genes (DEGs) between unsorted pseudoislets and native islets on days 0, 5, and 10 (FDR < 0.1; FC > 2 or FC < –2; *Supplementary file 4*). Differential expression analysis yielded 4, 299, and 137 DEGs on days 0, 5, and 10, respectively (*Figure 4c*). Not surprisingly, the gene expression profile of unsorted pseudoislets was almost identical to native islets on day 0 (*Figure 4d*). For example, we detected differences in expression levels between the groups for only four genes, namely, *CCN2* (cellular communication network factor 2), *DUSP5* (dual specificity phosphatase 5), *NR4A1* (nuclear receptor subfamily 4 group A member 1), *NR4A2* (nuclear receptor subfamily 4 group A member 2) on day 0 (*Figure 4d*). All four genes were significantly upregulated in unsorted pseudoislets which indicates that dissociation of native islets into single cells triggered acute changes in their expression. We detected changes in expression levels of 299 genes on day 5, among which 297 were downregulated and 2 (*GHRL, AC10875.5*) were upregulated in unsorted pseudoislets compared to native islets (*Figure 4e*). On day 10, all the 137 genes that were altered significantly in unsorted pseudoislets compared to native islets were downregulated (*Figure 4f*). Among these, 127 genes were consistently downregulated on day 5 as well as day 10. Pathway analysis of commonly downregulated genes on days 5 and 10 in the unsorted pseudoislets compared to native islets showed changes in pathways such as extracellular matrix (ECM) organization, integrin cell surface interactions, degradation of the ECM, complement system, and focal adhesion supporting the notion that the differences were likely due to physical separation of islets into single cells (*Figure 4g*, *Supplementary file 5*). Interestingly, we observed consistent downregulation of genes involved in the Hippo signaling pathway such as *TEAD2, YAP1, WWTR1, TGFB2, CCN2* on days 5 and 10. Expression of *CCN2* showed a dynamic change with upregulation on day 0 and downregulation on days 5 and 10 in unsorted pseudoislets compared to native islets.

## Time-dependent changes in the α-pseudoislet transcriptome

Next, to investigate whether the gene expression pattern of α-cells alters after separation from neighboring non-α islet cells, we explored genes and pathways that are progressively up- or downregulated in α-pseudoislets, unsorted pseudoislets, or native islets in culture over the period from day 0 to day 5 and day 5 to day 10. We identified 413 genes in α-pseudoislets, and 341 genes in unsorted pseudoislets, in contrast to only 11 genes in native islets that were significantly altered during this period (*Figure 5a*, *Supplementary file 6*). The fact that a majority of the genes (597 out of 608 genes) that were altered during this period in α-pseudoislets and unsorted pseudoislets but not in native islets likely reflects a transcriptional response of the cells following cell dissociation and re-aggregation. Pathway analyses of these 597 genes that were altered in α-pseudoislets or in unsorted pseudoislets but not in native islets revealed networks such as ECM organization, integrin cell surface interaction, focal adhesion, and collagen formation (*Figure 5b*, *Supplementary file 7*).

We then focused on the genes that were altered over the period of culture exclusively in α-pseudoislets. Among the 263 genes, 110 genes were upregulated and 153 genes were downregulated. We observed that genes including *MT1E* (metallothionein 1E), *MT1X* (metallothionein 1X), and *MT2A* (metallothionein 2A) belonging to the metallothionein gene family and the pathways such as response to metal ions, metallothioneins bind metals, and mineral absorption were upregulated in α-pseudoislets (*Figure 5c and d*, *Supplementary file 8*). Previous studies have identified variant alleles of metallothionein genes and their association with type 2 diabetes (*Yang et al., 2008*); however, their role in α-cells is not fully explored. Interestingly, pathways such as FoxO signaling, apoptosis, cell-ECM interactions, cGMP-PKG signaling, MAPK signaling, insulin secretion, and glucagon signaling pathway were all downregulated in the α-pseudoislets (*Figure 5c and d*). We observed that the β-cell enriched genes, *DLK1* (delta like non-canonical Notch ligand 1) also known as Pref-1, and *GSN* (gelsolin) were also downregulated significantly over time in α-pseudoislets (*Figure 4a*). On the other hand, *SMIM24* (small integral membrane protein 24), one of the α-cell enriched genes, was upregulated significantly in α-pseudoislets with time (*Figure 4a*). Using immunostaining, we validated that α-cells express SMIM24, GSN, and DLK1 and expression of these proteins altered

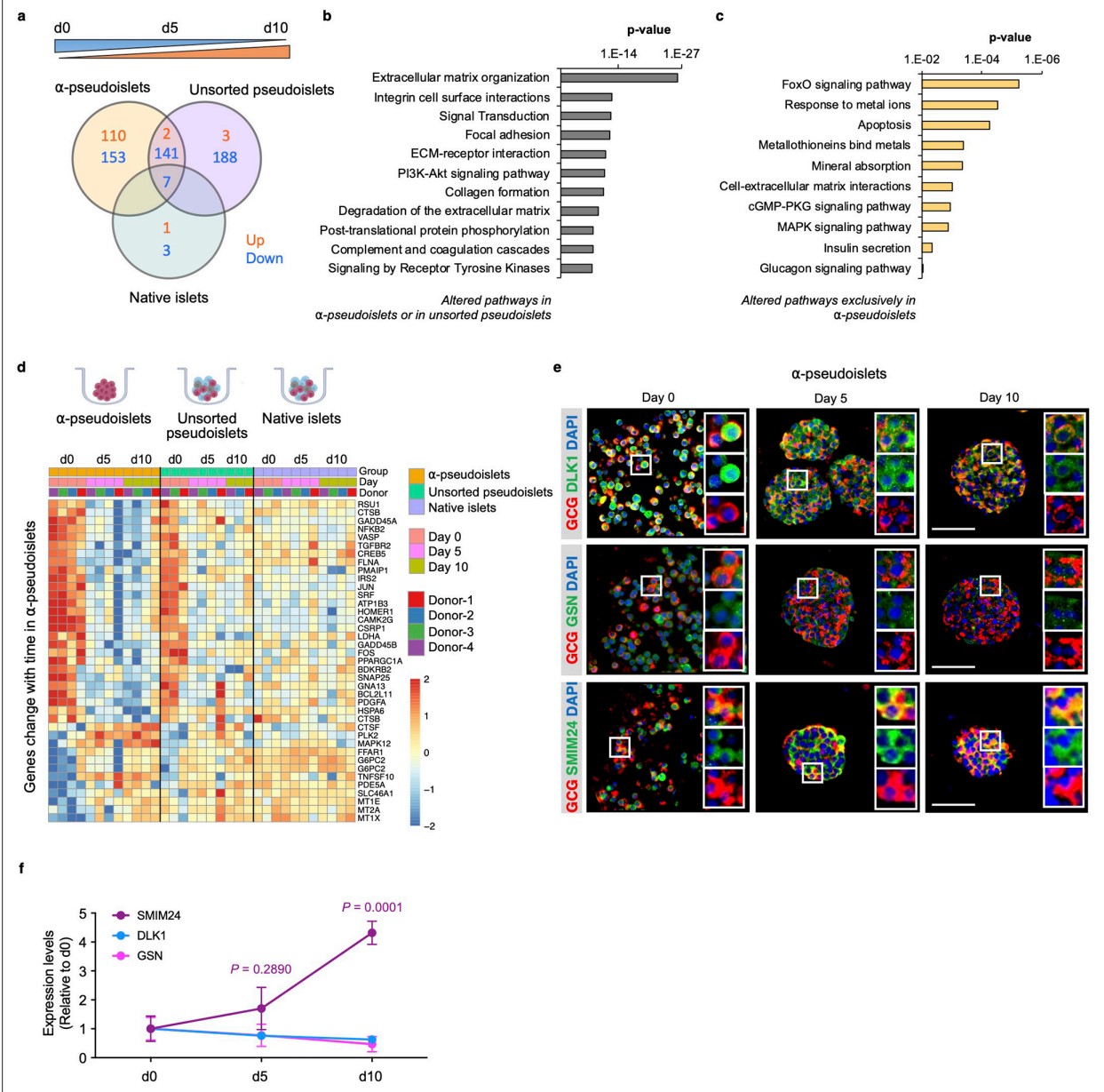

**Figure 5.** Time-dependent changes in transcriptome of α-pseudoislets. (**a**) Venn diagram shows number of genes that progressively up- or downregulated in α-pseudoislets, unsorted pseudoislets, and native islets in culture over the period from day 0 to day 5 to day 10. (**b, c**) Pathway analysis showing altered pathways in α-pseudoislets or in unsorted pseudoislets except native islets (**b**), and altered pathways only in α-pseudoislets with time (**c**). (**d**) Heatmap showing expression levels of genes significantly change with time only in α-pseudoislets. n=4 donors; α-pseudoislets d0, d5, d10, unsorted pseudoislets d5, native islets d5, d10, and n=3 donors; unsorted pseudoislets d0, d10, native islets d0 (**a–d**). (**e**) Representative immunostaining images of α-pseudoislets on day 0, 5, and 10 showing GCG (red), DLK1, GSN, and SMIM24 (green). Nuclei stained with DAPI are blue. Scale bar, 50 μm. (**f**) Expression level of each protein in α-pseudoislets on days 0, 5, 10. Data are presented as mean values ± s.e.m. Two-way ANOVA followed by Dunnett's multiple comparison test compared to d0. n=3 donors (**e, f**).

with time (*Figure 5e and f*). Whether changes in their expression levels are important for α-cells in the absence of cell-to-cell interactions or the paracrine signals derived from other islet cells require additional studies.

Overall, this study shows that human pancreatic α-cells can be purified by cell dissociation followed by DA-ZP1 labeling and FACS, and the purified α-cells can be maintained in culture to study secretory function, transcriptional changes, survival, and proliferation of α-cells.

## Discussion

Despite its importance in both physiology and pathophysiology, the progress in studies related to human α-cell biology has been relatively slower compared to β-cells. A major limitation is the technical challenge in obtaining adequate numbers of high-quality live human α-cells for analyses (*Haliyur et al., 2019*; *Xin et al., 2016*). In previous studies, live human α-cells were purified from human islet preparations by using a panel of cell-surface-binding monoclonal antibodies (*Dorrell et al., 2008*) or by combining antibody labeling approaches with ZIGIR labeling (*Ghazvini Zadeh et al., 2020*). Both methods enable isolation of live human α-cells with high purity, but require antibody staining, which increases the risk of cell loss during the labeling procedure and consequently limits α-cell yield. It has been reported that antibody-labeling approaches can recover 200,000 α-cells from 10,000 human islet equivalents (IEQ) (*Liu et al., 2019*). Here, we provide an antibody-free approach to obtain primary human α-cells using a fluorescein-tagged sensor which can recover ~2×-fold higher live α-cells with ~95% purity (*Figure 1—figure supplement 1*). Other advantages of the zinc-based labeling approach include: (1) the short incubation (10 min) for labeling with DA-ZP1 followed by sorting which reduces effort and limits stress on the cells; (2) zinc-based probes are cost-effective compared to antibodies and small quantities (80 nM) can be used for multiple experiments which allows scaling up of cell sorting processes. In regard to potential disadvantages is the ability of zinc-based probes to bind to intracellular Zn(II) which are important for cell functions including secretion of insulin. However, we have shown that DA-ZP1 does not affect insulin secretion, proliferation, or apoptosis in a human β-cell line (EndoC-βH1) even at high concentrations (10 μM) (*Kahraman et al., 2021*) and therefore this issue may not be a significant concern. Furthermore, since the levels of intracellular DA-ZP1 decrease gradually 24 hr after entering cells by passive diffusion and can last more than a week to disappear completely, further studies are warranted to determine additional effects of DA-ZP1. The function of re-aggregated α-pseudoislets was examined previously by Liu et al. after sorting individual islet cells by monoclonal surface antibodies (*Liu et al., 2019*). In their study, α-pseudoislets secreted glucagon at low concentrations of glucose but failed to respond to changes in glucose concentrations. They argued that the glucose-sensing machinery was deregulated in the absence of paracrine effects of neighboring β-cells. Conversely, our data demonstrated preserved glucose-sensing machinery and secretory actions of α-pseudoislets despite the absence of β-cells (*Figure 3*). While our data suggest that α-cells can regulate glucagon secretion by using cell intrinsic signaling in response to glucose in the absence of paracrine factors, such as insulin and somatostatin (*Elliott et al., 2015*) and juxtracrine-signaling factors, such as EphA4/7 on α-cells and ephrins on β-cells (*Reissaus and Piston, 2017*), it cannot exclude their contribution to secretory function in vivo. Our RNA-seq data supported maintenance of α-cell identity in α-pseudoislets (*Figure 4a and b*), and demonstrated culture time-dependent adaptive alterations in pathways involved in secretory capacity or glucagon signaling (*Figure 5c*). Thus, the discrepancy between the two studies could be explained by the use of different methods and time points in performing functional assays.

Human islets are micro-organs composed of multiple endocrine cell types and feature heterotypic cell-cell and cell-matrix interactions. Earlier studies demonstrated that paracrine communication between islet cells are important for their function and proliferation (*Gromada et al., 2018*). Additionally, vascular endothelial cells, neuronal projections, and ECM network within the islets also contribute regulatory signals to islet cells (*Aamodt and Powers, 2017*; *Lammert and Thorn, 2020*; *Ng et al., 2021*). This islet microenvironment is necessarily disrupted when islets are dissociated into single cells as in unsorted pseudoislets and likely absent in α-pseudoislets. The three groups, namely, (1) α-pseudoislets that are bereft of all other endocrine cells, (2) unsorted pseudoislets which include all endocrine cells but are distributed in a disorderly manner, and (3) native islets which consist of all endocrine cells with intact cell-cell interactions allowed us to dissect the effects of the microenvironment on transcriptional regulation of α-cells that reflects paracrine interactions. For example, over the 10 days in culture, we observed little change in the transcriptome of native islets compared with unsorted pseudoislets (*Figure 4b*, *Figure 5a*). Although the isolation procedure disrupts innervation and vasculature, the isolated islets preserve cell-cell and cell-matrix interactions. Almost no changes in the transcriptome of native islets in culture over the period of 10 days highlight the importance of intact islet structure on transcriptome stability.

Another example of the significance of the islet cell microenvironment is that genes involved in ECM organization and cell surface interactions were downregulated in unsorted pseudoislets compared

with native islets (**Figure 4g**). These observations are consistent with a previous report showing less abundant ECM components in re-aggregated pseudoislets than in intact islets (**Lorza-Gil et al., 2019**).

A notable finding revealed by RNA-seq data is that expression levels of α- or β-enriched genes were recovered after typical islet architecture was regained in unsorted pseudoislets (**Figure 4b**). These data support the fact that intact cell-cell and cell-matrix interactions are important to sustain cell identity in vitro (**Figure 4b**). The expression levels of α-cell enriched genes did not return toward normal in α-pseudoislets despite their ability to self-assemble and form tight clusters (**Figure 2—figure supplement 1**) which might argue that heterotypic cell-to-cell interactions are more important than homotypic interactions in maintaining α-cell identity. Indeed, previous studies reporting an upregulation of β-cell signature genes in α-cells after monotypic α-cell aggregation (**Furuyama et al., 2019**) or extreme β-cell loss (**Thorel et al., 2010**) support the relevance of heterotypic cell-to-cell interactions in maintenance of α-cell identity. The tighter cluster formation of α-pseudoislets was an interesting observation which could be due to preferential expression of adhesion proteins such as N-CAM (neural-cell adhesion molecule) (**Cirulli et al., 1994**). Overall, our RNA-seq data revealed important transcriptional effects secondary to dissociation of human islets and provide a useful resource for future studies on re-aggregated islet cells.

In conclusion, we report the use of a fluorescein-based sensor, to demonstrate isolation of live pancreatic α-cells with high purity and in vitro cultivation of α-pseudoislets without significantly affecting their identity and function over time. This unique tool can be used to isolate live human α-cells for functional studies, transcriptomic analyses, or studies assessing cellular toxicity or proliferation.

# Materials and methods

**Key resources table**

| Reagent type (species) or resource | Designation | Source or reference | Identifiers | Additional information |
|---|---|---|---|---|
| Antibody | Anti-C-peptide (Rat monoclonal) | Developmental Studies Hybridoma Bank | Cat# GN-ID4, RRID: AB_2255626 | IF (1:200) |
| Antibody | Anti-Ki67 (Mouse monoclonal) | BD Biosciences | Cat# BD550609, RRID: AB_393778 | IF (1:100) |
| Antibody | Anti-Insulin (Guinea Pig polyclonal) | Abcam | Cat# ab7842, RRID: AB_306130 | IF (1:400) |
| Antibody | Anti-Glucagon (Mouse monoclonal) | MilliporeSigma | Cat# G2654, RRID: AB_259852 | IF (1:10,000) |
| Antibody | Anti-Glucagon (Rabbit monoclonal) | Abcam | Cat# ab92517, RRID: AB_10561971 | IF (1:5000) |
| Antibody | Anti-DLK1 (Rabbit polyclonal) | Abcam | Cat# ab21682, RRID: AB_731965 | IF (1:1000) |
| Antibody | Anti-GSN (Rabbit polyclonal) | MilliporeSigma | Cat# HPA054026, RRID: AB_2682347 | IF (1:50) |
| Antibody | Anti-SMIM24 (Rabbit polyclonal) | MilliporeSigma | Cat# HPA045046, RRID: AB_10964444 | IF (1:50) |
| Antibody | Anti-Guinea Pig AF594 (Donkey polyclonal) | Jackson ImmunoResearch | Cat# 706-586-148, RRID: AB_2340475 | IF (1:400) |
| Antibody | Anti-Rat AF488 (Donkey polyclonal) | Jackson ImmunoResearch | Cat# 712-546-153, RRID: AB_2340686 | IF (1:400) |
| Antibody | Anti-Mouse AF594 (Donkey polyclonal) | Jackson ImmunoResearch | Cat# 715-586-150, RRID: AB_2340857 | IF (1:400) |
| Antibody | Anti-Mouse AF488 (Donkey polyclonal) | Jackson ImmunoResearch | Cat# 715-545-150, RRID: AB_2340846 | IF (1:400) |
| Antibody | Anti-Rabbit AF594 (Donkey polyclonal) | Jackson ImmunoResearch | Cat# 711-586-152, RRID: AB_2340622 | IF (1:400) |
| Antibody | Anti-Rabbit AF488 (Donkey polyclonal) | Jackson ImmunoResearch | Cat# 711-545-152, RRID: AB_2313584 | IF (1:400) |

*Continued on next page*

*Continued*

| Reagent type (species) or resource | Designation | Source or reference | Identifiers | Additional information |
|---|---|---|---|---|
| Biological sample (*Homo sapiens*) | Primary human pancreatic islets | Integrated Islet Distribution Program (IIDP), Prodo Laboratories Inc, ADI Islet Core | http://iidp.coh.org; RRID: SCR_014387, https://prodolabs.com/ | Freshly isolated |
| Chemical compound | DA-ZP1 | Laboratory of Amit Choudhary | Broad Institute of MIT and Harvard | |
| Chemical compound | ZIGIR | Laboratory of Wen-hong Li | University of Texas, Dallas | |
| Chemical compound | NPG | Thermo | N7991 | |
| Other | DAPI, dilactate | Sigma | D9564 | IF (1:6600) |
| Other | Miami Media #1A | Cellgro | 98-021-CV | Islet cell culture |
| Other | TrypLE | Thermo Fisher Scientific | 12604-013 | Islet cell dissociation |
| Other | FBS | Thermo Fisher Scientific | 10437028 | Islet cell dissociation |
| Other | DPBS | Thermo Fisher Scientific | 14190250 | Islet cell dissociation |
| Other | D-Glucose | Sigma-Aldrich | G8769 | Glucagon secretion assay |
| Other | FFA-BSA | Sigma-Aldrich | 3117057001 | Glucagon secretion assay |
| Other | DMSO | Sigma-Aldrich | D2650-100 | FACS |
| Other | Penicillin-streptomycin | Corning | 30-002-Cl | Islet cell culture |
| Other | 4% PFA | Wako | 163-20145 | Embedding islets in agar |
| Other | Antibody diluent | Abcam | Ab64211 | IF |
| Other | Low melting agarose | Scientific Laboratory Supplies | NAT1030 | Embedding islets in agar |
| Other | TRIzol reagent | Invitrogen | 15596026 | RNA isolation |
| Commercial assay or kit | Glucagon ELISA kit | Mercodia | 10-1271-01 | |
| Commercial assay or kit | MycoAlert Mycoplasma Test Kit | Lonza | LT07-318 | |
| Commercial assay or kit | CellTiter-Glo Luminescent Cell Viability Assay Kit | Promega | G7570 | |
| Commercial assay or kit | ApopTag Peroxidase In Situ Apoptosis Detection Kit | MilliporeSigma | S7100: RRID:AB_2661855 | |
| Commercial assay or kit | QIAGEN RNeasy micro kit | QIAGEN | 74004 | |
| Software, algorithm | Prism v.7.0 | GraphPad Software | http://www.graphpad.com; RRID: SCR_002798 | |
| Software, algorithm | Flowjo-v10 | FlowJo Software | http://www.flowjo.com; RRID: SCR_008520 | |
| Software, algorithm | ImageJ | ImageJ Software | https://imagej.net: RRID:SCR_003070 | |
| Software, algorithm | R version 4.1.0 | R Software | https://www.r-project.org/ | |

## Primary human cells

Human islets were obtained from the Integrated Islet Distribution Program (IIDP) and Prodo Labs. Upon receipt, islets were centrifuged at 200 × *g* for 1 min and resuspended in fresh Miami medium no. 1A (Cellgro). Cells were transferred to Petri dishes and cultured in 5% $CO_2$ at 37°C overnight before performing the FACS experiments. The donor demographic information is summarized in

*Supplementary file 9*. All studies and protocols used were approved by the Joslin Diabetes Center's Committee on Human Studies (CHS no. 5-05).

## Flow cytometry and islet cell culture

Twenty-four hours after receiving the human islets (day 0), 1000 islets were handpicked and transferred to 10 round-bottom non-treated 96-well plates (Corning #3788) (1 islet/well in 200 µl Miami medium) and kept in culture. The rest of the human islets, approximately 15,000 IEQ, were collected in a 15 ml tube, washed with DPBS, and resuspended in 3 ml of TrypLE for single-cell dispersion (*Lee et al., 2020*). The islets were dispersed into a single-cell suspension in TrypLE for 12–15 min at 37°C. Dissociated islet cells were washed with DMEM medium containing 10% FBS and resuspended in the Miami medium containing DA-ZP1 (80 nM) for 10 min at 37°C. The cells were then filtered through a 30 µm filter to remove any aggregates before FACS sorting. Approximately 1/10th of the DA-ZP1-treated cell suspension were labeled as 'unsorted islet cells' and set aside while the rest of the DA-ZP1-treated cells was sorted by FACSAria cell sorter (BD Biosciences, Joslin Flow Cytometry Core). After FACS, both sorted and unsorted islet cells were centrifuged at 250 × *g* for 5 min, resuspended in Miami media, counted using 0.4% trypan blue stain by cell counter (Nexcelom Bioscience) to determine number of live cells, and immediately seeded in round-bottom non-treated 96-well plates (Corning #3788) (1000 cells/200 µL/well or 2000 cells/200 µL/well). The day of sorting is considered 'day 0'. Half of the medium was refreshed every other day. Analysis of flow cytometry data was completed using FlowJo 10.7.1 (FlowJo LLC, Ashland, OR, USA). The gating strategy is shown in *Figure 1—figure supplement 1*. DA-ZP1 is synthesized by Amit Choudhary's lab. Compound structure and synthesis of DA-ZP1 are provided in *Lee et al., 2020*; *Kahraman et al., 2021*. One mM of ZIGIR (kindly provided by Dr. Wen-hong Li; University of Texas, Dallas, TX, USA) was dissolved in DMSO and used at a final concentration of 1 µM. One mg Newport Green DCF (Thermo Fisher, N7991) was solubilized in DMSO and used at a final concentration of 1 µM. Cells treated with DA-ZP1 (ex/em 495/500–650), ZIGIR (ex/em 571/543), and NPG (ex/em 485/530) were sorted by the FACSAria cell sorter.

## Immunocytochemistry

To profile endocrine cell types after sorting, both sorted and unsorted islet cells were seeded in Matrigel-coated flat-bottom 96-well plates (1000 cells/200 µL/well) immediately after FACS. Next day (day 1), the cells were fixed in 4% PFA (Wako) for 15 min at room temperature and washed with PBS. Cells were then permeabilized and blocked with PBS containing 0.25% Triton-X and 5% donkey serum (Sigma) for 1 hr at room temperature. Primary antibodies C-peptide (DSHB, 1:200) and glucagon (MilliporeSigma, G2654, 1:10,000) diluted in antibody dilution buffer (Abcam) were added to the wells for overnight at 4°C. Cells were washed three times with PBS and the secondary antibody, diluted in PBS, was added to the wells for 1 hr at room temperature. Cells were washed three times with PBS and DAPI (Sigma) was added to the wells. Images were captured using an Olympus IX51 Inverted Microscope. For estimation of cell composition, ~4000 cells were counted per donor, and data were expressed as percentage of hormone+ cells.

## Cell viability assay

The assay was performed using a Cell Titer Glo luminescence cell viability kit according to the manufacturer's instructions on days 5 and 10 post-sorting. Approximately 8000 cells (eight α-pseudoislets, eight unsorted pseudoislets, and eight native islets) were added to the wells of opaque-walled 96-well plates and 100 µl of Cell Titer Glo reagent was added to the cells. The content was mixed for 2 min on an orbital shaker and incubated for 10 min at room temperature. Luminescence was recorded using a Promega GloMax luminometer with an integration time of 0.3 s per well.

## Islet immunohistochemistry and quantification

Sorted or unsorted islet cells, and native islets seeded in round-bottom non-treated 96-well plates were collected in a 15 ml tube (~100 wells/tube) on days 0, 5, or 10 post-sorting, washed with PBS, and fixed in 4% PFA for 15 min at room temperature. The cells were washed, embedded in agarose and paraffin, sectioned and used for immunostaining. Sections were stained using antibodies against Ki67 (BD550609, 1:100), insulin (Abcam, ab7842, 1:400), glucagon (MilliporeSigma, G2654, 1:10,000 or

Abcam, ab92517, 1:5000) DLK1 (Abcam, ab21682, 1:1000), GSN (Sigma, HPA054026, 1:50), SMIM24 (Sigma, HPA045046, 1:50), and TUNEL (ApopTag, Chemicon, S7100) and counterstained with DAPI (MilliporeSigma, D9564, 1:6600). For estimation of cell composition, ~4000 cells were counted per donor and data were expressed as percentage of hormone+ cells. For estimation of cell proliferation, ~3000 GCG+ cells were counted per donor and data were expressed as percentage of Ki67+GCG+ cells. To assess cell death, ~1000 GCG+ cells were counted per donor and apoptotic index was measured by quantification of the percentage of TUNEL+GCG+ cells. Expression levels of DLK1, GSN, and SMIM24 in islet sections were measured using ImageJ. Fluorescent images were captured using a Zeiss Axio Imager A2 upright fluorescence microscope using the same exposure time. The mean fluorescence intensity (MFI) was quantified in the selected islet area and the mean fluorescence of background is subtracted from the MFI to find corrected total cell fluorescence. Approximately 50 islets were analyzed per islet donor, and data were expressed as relative expression levels compared to day 0.

## Secretion assay

On day 5 post-sorting, eight α-pseudoislets, eight unsorted pseudoislets, and eight native islets were transferred to wells of a U-bottom non-treated 96-well plate (Corning #3788), and preincubated in Krebs-Ringer bicarbonate (KRB) buffer containing 135 mmol/L NaCl, 3.6 mmol/L KCl, 5 mmol/L NaHCO$_3$, 0.5 mmol/L NaH$_2$PO$_4$, 0.5 mmol/L MgCl$_2$, 1.5 mmol/L CaCl$_2$, 10 mmol/L HEPES, pH 7.4, 0.1% FFA-BSA with 16.7 mM glucose for an hour. Static glucose challenge was then initiated by adding KRB buffer containing 3.3 mM or 16.7 mM glucose for 1 hr. Aliquots of supernatants were removed for later analysis and ice-cold acid ethanol was added to extract the glucagon content from the cells. Glucagon release and content were measured by the human glucagon ELISA (Mercodia) according to the manufacturer's instructions.

## RNA isolation, sequencing, and data analysis

Approximately 100 α-pseudoislets, 100 unsorted pseudoislets, and 100 native islets were lysed in TRIzol reagent (Invitrogen) according to the manufacturer's instructions and the resultant aqueous phase was mixed (1:1) with 70% RNA-free ethanol. The mixture was added to QIAGEN RNeasy micro kit columns and total RNA was extracted following the manufacturer's protocols. Genomic DNA was digested using RNase-Free DNase kit (QIAGEN). The RNA quality and quantity were analyzed using a NanoDrop 1000 Spectrophotometer (Thermo Fisher) and library was constructed using Takara Pico-Input Strand-Specific Total RNA-seq for Illumina (Takara). RNA-seq was performed on an Illumina NovaSeq 6000 according to the manufacturer's instructions. Approximately 50 million paired-end 100 bp reads were generated for each sample. We aligned the adapter-trimmed reads to the human transcriptome using Kallisto, converted transcript counts to gene counts using tximport, normalized the counts by trimmed mean of M-values (*Robinson and Oshlack, 2010*), and transformed normalized counts into log2 counts per million with Voom (*Law et al., 2014*). We applied ComBat-Seq (*Zhang et al., 2020*) to remove the effect of known batches, and then assessed genes' association with time and differential expression between groups using the linear regression modeling package limma (*Ritchie et al., 2015*). We corrected for testing many genes with the FDR. R version 4.1.0 was used. Pathway analysis was done using the ConsensusPathDB interaction database (http://cpdb.molgen. mpg.de/CPDB).

## scRNA-seq analysis of GSE84133

We downloaded this previously published dataset (*Baron et al., 2016*) from the Gene Expression Omnibus. We filtered out cells that have less than 2000 total gene counts and 1000 detected genes, and removed genes that have average counts of 0.01 or less. Similar cells were clustered together using a graph-based clustering algorithm and the data was then normalized (*Lun et al., 2016*). Moderated t-tests from the linear regression modeling R package limma (*Ritchie et al., 2015*) were performed to detect genes that are differentially expressed between β- and α-cells, with subject and cellular detection rate (i.e. the fraction of detected genes) as covariates.

## Statistical analysis

All statistics were performed using GraphPad Prism software version 7.0a (GraphPad Software Inc, La Jolla, CA, USA). Specific statistical tests for each experiment are described in the figure legends.

## Study approval

All human studies and protocols used were approved by the Joslin Diabetes Center Committee on Human Studies (CHS, 5-05). Formal consent from human islet donors was not required because samples were discarded islets from de-identified humans.

## Acknowledgements

We thank Hui Pan and Jonathan Dreyfuss (Joslin Bioinformatics and Biostatistics) for analyzing RNA-seq data, Natalie K Brown (Joslin) and Oluwaseun Ijaduola (Joslin) for fluorescence microscopy, Alison Marotta (Joslin) and Angela Wood (Joslin) for their assistance with FACS. Flow cytometry experiments were performed in the Joslin Flow Core, supported by the DRC (P30DK036836 and S10 OD021740-01). Human islets obtained from IIDP, supported by NIH (2UC4DK098085), and Prodo Labs. Funding: This work was supported by U01 DK123717 (to RNK and BW), UC4 DK116255 (to RNK, AC, and BW), R01 067536 (to RNK), U01 DK137242 (to AC and RNK), and R01 DK132900 (to AC and RNK).

## Additional information

### Competing interests

Sevim Kahraman: is an employee of Boehringer Ingelheim Pharmaceuticals, Inc. Rohit N Kulkarni: is on the Scientific Advisory Board of Novo Nordisk, Biomea and Inversago Therapeutics. The other authors declare that no competing interests exist.

### Funding

| Funder | Grant reference number | Author |
| --- | --- | --- |
| National Institutes of Health | U01 DK123717 | Bridget Wagner<br>Rohit N Kulkarni |
| National Institutes of Health | UC4 DK116255 | Bridget Wagner<br>Rohit N Kulkarni<br>Amit Choudhary |
| National Institutes of Health | R01 067536 | Rohit N Kulkarni |
| National Institutes of Health | U01 DK137242 | Amit Choudhary<br>Rohit N Kulkarni |
| National Institutes of Health | R01 DK132900 | Amit Choudhary<br>Rohit N Kulkarni |

The funders had no role in study design, data collection and interpretation, or the decision to submit the work for publication.

### Author contributions

Sevim Kahraman, Conceptualization, Data curation, Formal analysis, Supervision, Validation, Investigation, Visualization, Methodology, Writing – original draft, Project administration, Writing – review and editing; Kimitaka Shibue, Data curation, Formal analysis, Validation, Investigation, Visualization, Methodology, Writing – review and editing; Dario F De Jesus, Data curation, Formal analysis, Investigation, Methodology, Writing – review and editing; Hyunki Kim, Data curation, Formal analysis, Validation, Investigation, Methodology, Writing – review and editing; Jiang Hu, Resources, Data curation, Formal analysis, Investigation, Methodology, Writing – review and editing; Debasish Manna, Resources, Investigation, Methodology, Writing – review and editing; Bridget Wagner, Amit Choudhary, Conceptualization, Resources, Funding acquisition, Writing – review and editing; Rohit

N Kulkarni, Conceptualization, Resources, Supervision, Funding acquisition, Investigation, Project administration, Writing – review and editing

### Author ORCIDs
Sevim Kahraman https://orcid.org/0000-0002-2880-6589
Rohit N Kulkarni https://orcid.org/0000-0001-5029-6119

### Ethics
All human studies and protocols used were approved by the Joslin Diabetes Center Committee on Human Studies (CHS, 5–-05). Formal consent from human islet donors was not required because samples were discarded islets from de-identified humans.

### Decision letter and Author response
Decision letter https://doi.org/10.7554/eLife.85056.sa1
Author response https://doi.org/10.7554/eLife.85056.sa2

---

## Additional files

### Supplementary files
• Supplementary file 1. Single-cell RNA-sequencing (scRNA-seq) analysis of GSE84133 dataset revealed genes that are differentially expressed between islet α-cells and β-cells.

• Supplementary file 2. Changes in expression levels of α-cell enriched and β-cell enriched genes in sorted α-cells on day 0.

• Supplementary file 3. Changes in expression levels of α-cell enriched and β-cell enriched genes in α-pseudoislets, unsorted pseudoislets, and native islets on day 5 and day 10 vs day 0.

• Supplementary file 4. Differentially expressed genes between unsorted pseudoislets versus native islets with fold change >2 and <−2 (log2 FC >1 and <−1) and false discovery rate (FDR)<0.1.

• Supplementary file 5. Pathway analysis of commonly downregulated genes on days 5 and 10 in the unsorted pseudoislets compared to native islets.

• Supplementary file 6. Progressively up- or downregulated genes over the period from day 0 to day 10.

• Supplementary file 7. Altered pathways in α-pseudoislets and unsorted pseudoislets except native islets with time.

• Supplementary file 8. Altered pathways exclusively in α-pseudoislets with time.

• Supplementary file 9. Donor information.

• MDAR checklist

### Data availability
RNA-seq data have been deposited under accession code GSE199412. Further information and requests for resources and reagents should be directed to the corresponding author.

The following dataset was generated:

| Author(s) | Year | Dataset title | Dataset URL | Database and Identifier |
|---|---|---|---|---|
| Kahraman S, Kulkarni RN | 2023 | Using Zinc-based probes to generate pseudoislets from purified human alpha-cells for functional analyses | https://www.ncbi.nlm.nih.gov/geo/query/acc.cgi?acc=GSE199412 | NCBI Gene Expression Omnibus, GSE199412 |

The following previously published dataset was used:

| Author(s) | Year | Dataset title | Dataset URL | Database and Identifier |
|---|---|---|---|---|
| Veres A, Baron M | 2016 | A single-cell transcriptomic map of the human and mouse pancreas reveals inter- and intra-cell population structure | https://www.ncbi.nlm.nih.gov/geo/query/acc.cgi?acc=GSE84133 | NCBI Gene Expression Omnibus, GSE84133 |

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
