## [Editor Report]

The manuscript by Kahraman et al. describes the use of the fluorogenic dye – diacetylated Zinpyr1 (DA-ZP1) – for purifying cadaveric human islet α cells. The data show that DA-ZP1 is a useful non-antibody-based approach to label α cells and provides additional evidence that the purified α cells remained viable and functional after several days in culture. This resource will be a useful tool for islet biologists and researchers investigating islet cell dysfunction in diabetes.

---

## [Decision Letter]

**Decision letter after peer review:**

Thank you for submitting your article "Fluorescein-based sensors to purify human α-cells for functional and transcriptomic analyses" for consideration by *eLife*. Your article has been reviewed by 3 peer reviewers, one of whom is a member of our Board of Reviewing Editors, and the evaluation has been overseen by David James as the Senior Editor. The following individual involved in review of your submission has agreed to reveal their identity: Peter Thompson (Reviewer #2).

Essential revisions:

1) Increase the number of biological replicates for the functional analyses.

2) Provide comparisons with other documented techniques or more thoroughly assess and report the purity of the sorted populations.

3) Provide additional experimental details and clarifications as outlined by Reviewer 3.

*Reviewer #1 (Recommendations for the authors):*

1. Perform RNA-Seq on sorted high and medium intensity populations to more accurately and comprehensively assess the amount of cross contamination with other cell populations, including β cells, δ cells and non endocrine populations.

2. Directly compare the new method with other published methods of sorting human α cells to show the advantages and disadvantages of the different methods.

3. Increase the n's for the functional analysis of the α pseudo islets – especially since the secretion is very low (although statistically significant) and the author's results contradict Liu et al., 2019.

4. Indicate how the glucagon content in figure 3d is normalized. Were equal numbers of α cells used in the unsorted and whole islets compared to the α-pseudo islets?

*Reviewer #2 (Recommendations for the authors):*

Additional experiments. Some additional experiments that could strengthen the manuscript, its significance and conclusions are:

– A side-by-side comparison of DA-ZP1 alongside one other Zn-based probe and how it compares in terms of labeling efficiency, fluorescence intensity, background etc for α cells.

­ – Additional glucagon secretion assays with at least one or two more human islet donor preparations would strengthen the generalizability of the conclusions around the pseudoislet data.

*Reviewer #3 (Recommendations for the authors):*

1) Terminology: Please provide further rationale for referral to α cells as "intermediate" in this manuscript as opposed to "negative" in previous paper in JACS.

2) Figures 1d, 2b-j, 3b-d, 5e-f: The figure legends state values represent "biological replicates using islet cells from a single donor". Further clarification is needed. It is unclear whether these are actually technical replicates, and if so, data would be greatly strengthened by providing corresponding data from additional 3 donors analyzed and displaying values in aggregate.

3) Figure 2b: Why are pseudoislets prepared from unsorted islet cells 50-60% larger than those prepared from purified α cells, given the same number of cells are seeded and there is no difference in viability in 2c?

4) Figure 2: Cell viability is not significantly different between α-pseudoislets and unsorted pseudoislets in 2d, yet the apoptotic index is much lower in α-pseudoislets in 2h. What could explain these differences?

5) Figure 3: What day of reaggregation/in culture were pseudoislets and native islets used for these experiments?

6) Figure 3c-d: Analyzing data using a one-way ANOVA would be more appropriate as opposed to multiple t-tests.

7) Figure 4, Supplementary File 1: Do the genes classified as α- and β-enriched replicate across multiple single cell RNA-seq data sets?

8) Figure 4a: Does this mean that native islets/pseudoislets from the same donor were followed longitudinally – day 0, day 5, and day 10?

9) Lines 128-129: This sentence is difficult to understand. Does this mean that the gene expression in α cell-enriched pseudoislets returned to normal after 10 days of culture? Or genes differentially expressed in α cells at 5 and 10 days, were "unchanged" in pseudoislets made without cell sorting as well as whole islets?

10) Figure 5b-d: Are these paired experiments comparing α cell-enriched pseudoislets with pseudoislets made of unsorted islet cells and native islets? This is very difficult to follow because the proportion of cell types has changed after α cell purification.

11) Figure 4d: Does this imply dispersion doesn't influence gene expression?

12) Figures 4-5: The authors utilize the terms "day 0", "day 5", and "day 10", although the meaning of these terms is not unclear. Is this in reference to days post-sort or the number of days in culture after the pseudoislets have formed? If the former, clarity overall would be improved by referring to cells as "sorted α cells" and "unsorted islet cells" prior to pseudoislet formation. After reaggregation, referral to these as "α -pseudoislets", "unsorted pseudoislets", and "native islets" would be more appropriate. For whole islets, does "day 0" imply day of arrival?

13) Figure 2b: Is the time scale here (beginning with Day 1) have the same meaning as the time points in Figures 4 and 5?

14) Figures 4a,5d: The color choice used to signify each donor makes the figure difficult to follow. More contrast between each assigned color would make it clearer.

15) Lines 522-523: What is the magnification/scale bar size for the images in the top row of 2e?

16) Figure 5d: DLK1, GSN, and SMIN24 do not appear in the heatmap, though emphasized in the text.

17) Figure 5e: Order of insets for GCG and SMIM24 staining at Day 0 are switched compared to other panels.

18) Lines 163-167: Are genes confirmed to also be downregulated at day 5, or was comparison only made between day 0 and day 10?

19) Lines 581-584: The stated description of data included in the heatmap is difficult to understand.

20) Supplementary Figure 1c: Please provide the number of live cell pre-FACS as well as the number of live α cells recovered.

21) Supplementary Files 2,3, and 6 – More detailed data (reads, log2FC, FDR) for genes reported is required; listing "up" or "down" is unclear.

22) Methods: Was viability PI/DAPI stain used during the sort to gate on live cells (Supplementary Figure 1a)?

23) Methods/Materials: Include catalog number for U-bottom plates used for pseudoislets.

24) Methods: Are islet preps handpicked before sorting?

25) Methods: How many cells were counted to measure composition of α pseudoislets, unsorted pseudoislets, and whole islets (Figure 2f)?

26) Methods: Are "unsorted" cells still labeled with DA-ZP1? If not, please discuss potential caveats of comparing unlabeled, unsorted cells with labeled cells.

27) Methods: Please provide further detail on how expression level in 5f is calculated from images in 5e.

28) Discussion: Please include a discussion of potential limitations to using this probe over antibody-based sorting approaches.

29) References 21 and 22 are duplicated.

---

## [Author Response]

Essential revisions:1) Increase the number of biological replicates for the functional analyses

We have tested an additional donor cells to increase the n number of the functional assay (new Figure 3).

2) Provide comparisons with other documented techniques or more thoroughly assess and report the purity of the sorted populations

We have performed additional experiments and compared DA-ZP1 with other Zinc binding molecules (Figure 1g) and report the purity of DA-ZP1-mediated sorted population by comparing transcriptomics of the low, intermediate, bright subsets and unsorted cells in Figure 1f.

3) Provide additional experimental details and clarifications as outlined by Reviewer 3.

We have added experimental details and clarified the points raised by Reviewer 3.

Reviewer #1 (Recommendations for the authors):1. Perform RNA-Seq on sorted high and medium intensity populations to more accurately and comprehensively assess the amount of cross contamination with other cell populations, including β cells, δ cells and non endocrine populations.

We have compared transcriptomics of DA-ZP1 low, intermediate, bright, and unsorted populations and observed expression of α, β, other endocrine, and non-endocrine cell markers in different populations to assess cross contamination using 3 islet donors. The results are in a new Figure 1f.

2. Directly compare the new method with other published methods of sorting human α cells to show the advantages and disadvantages of the different methods.

We compared the new method with other zinc binding molecules such as NPG (Newport Green) (Kirkpatrick et al., 2010, PMID: 20548773) and ZIGIR (zinc granule indicator) (Ghazvini Zadeh et al., 2020, PMID: 32668245) reported previously to purify live human/murine α cells. NPG-mediated α cell sorting enabled isolation of human α cells with 74.5% purity (contamination with 6% β cells and 19% other cells) and ZIGIR-mediated α cell sorting enabled isolation of > 95% pure α cells when used in combination with the HPi2 antibody and TM4SF4 antibody. Although NPG was not as effective as DA-ZP1 in sorting α cells (74.5% α cell purity with NPG vs > 95% α cell purity in DA-ZP1), ZIGIR was highly effective in purifying α cells when combined with HPi2 antibody labeling endocrine cells (Dorell et al., 2008, PMID: 19383399) and TM4SF4 antibody for tetraspanin protein functioning as a marker to enrich for α cells (Muraro et al., 2016, PMID: 27693023). To perform a side-by-side comparison of DA-ZP1 with ZIGIR or NPG, we used these Zinc-binding molecules without assistance from antibodies during the staining of islet cells. FACS analysis showed that both ZIGIR-stained cells and NPG-stained cells generated a single bright subset with no distinct intermediate cell population as opposed to what we observed with DA-ZP1 staining. One could draw a gate to divide the bright population in two subsets (low and high subsets) to try to separate α and β cells, but this could result in variation in purity between different sorting and also affect the cell yield since α and β cells are not clearly separated with ZIGIR or NPG labeling without antibodies. The results are presented in the new Figure 1g.

3. Increase the n's for the functional analysis of the α pseudo islets – especially since the secretion is very low (although statistically significant) and the author's results contradict Liu et al., 2019.

We have performed functional analysis on an additional donor and presented data in new Figure 3.

4. Indicate how the glucagon content in figure 3d is normalized. Were equal numbers of α cells used in the unsorted and whole islets compared to the α-pseudo islets?

The data in Figure 3d is not normalized since we used equal numbers of ⍺-pseudoislets, unsorted pseudoislets, and native islets.

Reviewer #2 (Recommendations for the authors):Additional experiments. Some additional experiments that could strengthen the manuscript, its significance and conclusions are:– A side-by-side comparison of DA-ZP1 alongside one other Zn-based probe and how it compares in terms of labeling efficiency, fluorescence intensity, background etc for α cells.

We have performed a side-by-side comparison. Please see comments under Reviewer 1 (#2).

­ – Additional glucagon secretion assays with at least one or two more human islet donor preparations would strengthen the generalizability of the conclusions around the pseudoislet data.

We have performed functional analysis on additional islet donor and presented in new Figure 3.

Reviewer #3 (Recommendations for the authors):1) Terminology: Please provide further rationale for referral to α cells as "intermediate" in this manuscript as opposed to "negative" in previous paper in JACS.

Introduction includes more clarity on gating strategy used in the previous vs current paper.

2) Figures 1d, 2b-j, 3b-d, 5e-f: The figure legends state values represent "biological replicates using islet cells from a single donor". Further clarification is needed. It is unclear whether these are actually technical replicates, and if so, data would be greatly strengthened by providing corresponding data from additional 3 donors analyzed and displaying values in aggregate.

Only the data shown in Figure 2j are from n=3 donors and the data given in Figure 1d,e, 2b-h, 3b-d, 5e-f are from a single donor. We analyzed 3 different tubes, wells, or tissue sections of islets from a single donor in the first manuscript to generate the data. Although the islets were obtained from the same donor, we grouped islets in 3 different wells, tubes, or sections and considered them as three different biological replicates given the fact that human pancreatic islets are heterogenous in nature and have different compositions, sizes, and structures. Upon your request, we have analyzed islets obtained from additional donors as shown in Figure 1d,e, 2b-h, 3b-d, 5e-f. Figure legends have been updated.

3) Figure 2b: Why are pseudoislets prepared from unsorted islet cells 50-60% larger than those prepared from purified α cells, given the same number of cells are seeded and there is no difference in viability in 2c?

We counted the number of live cells after FACS and plated the same number of cells in each well. Next day after sorting (on day 1), sorted ⍺-cells and unsorted cells had similar sizes (Figure 2b vs 2c). Interestingly, on day 3, the sorted ⍺-cells already formed tight clusters while unsorted islet cells were still in the process of aggregation and appeared as disorganized cell clusters. The reason for this could be preferential expression of adhesion proteins such as N-CAM in ⍺-cells and promoting the aggregation by providing connections between ⍺-cells. This possibility is discussed and we provide representative data in Figure 2—figure supplement 1 comparing cluster sizes on day 5 in ⍺-pseudoislets vs unsorted pseudoislets.

Another possibility is that FACS could make sorted cells more fragile in the first days of cell culture compared to unsorted cells. We have not checked the changes in viability of sorted or unsorted cells on day 1 or 2. So, we speculate that more sorted ⍺-cells died shortly after plating compared to unsorted islet cells in the early days. However, luminescence levels which is an indication of the number of viable cells in culture based on quantitation of the ATP present, are very similar in ⍺-pseudoislets and unsorted pseudoislets on day 5 or day 10 (Figure 2d). So, this is probably not the sole reason of size difference between ⍺-pseudoislets and unsorted pseudoislets.

4) Figure 2: Cell viability is not significantly different between α-pseudoislets and unsorted pseudoislets in 2d, yet the apoptotic index is much lower in α-pseudoislets in 2h. What could explain these differences?

The reason for using cell viability assay was to determine the overall health of cells in culture by measuring ATP levels. Higher luminescence (ATP) levels indicated higher number of living cells. One drawback of this assay could be that measuring ATP may not be sensitive to capture changes at single cell level. In other words, a higher apoptotic index detected by TUNEL assay at single cell level in unsorted pseudoislets compared to α-pseudoislets in the previous version of this manuscript might not be captured by ATP level measurements.

Additionally, the data in Figure 2d in the first version of the manuscript expressed luminescence levels relative to day 5 of each group to show whether culturing cells up to 10 days affect viability. Now in the current version, we have tested another donor and presented the raw luminescence levels (normalized to blank control) to allow comparing viability of α-pseudoislets with other groups on day 5 and day 10. The new data is consistent with the previous data showing that viability of α-pseudoislets is not significantly different compared to unsorted pseudoislets. Only native islets showed decreased viability on day 10 vs day 5 possibly due to hypoxia in the islet core. We also increased the number of donors for the TUNEL assay. The new data (Figure 2h) does not show significant changes in apoptotic index between α-pseudoislets and unsorted pseudoislets. We only detected a small but insignificant increase in apoptotic index in native islet on day 10 versus day 5.

5) Figure 3: What day of reaggregation/in culture were pseudoislets and native islets used for these experiments?

Functional assay was performed on day 5 post-sorting and is mentioned in Methods and Figure 3a legend.

6) Figure 3c-d: Analyzing data using a one-way ANOVA would be more appropriate as opposed to multiple t-tests.

The functional assay has been performed in an additional donor. The new data have been analyzed using one-way ANOVA and Figure 3 has been updated.

7) Figure 4, Supplementary File 1: Do the genes classified as α- and β-enriched replicate across multiple single cell RNA-seq data sets?

We used a single scRNA-seq dataset from four human donors (GSE84133) to identify α-cell enriched and β-cell enriched genes similar to other studies published previously (PMID: 31068696).

8) Figure 4a: Does this mean that native islets/pseudoislets from the same donor were followed longitudinally – day 0, day 5, and day 10?

Yes, ⍺-pseudoislets, unsorted pseudoislets, and native islets from the same donor were followed longitudinally. Donor information is provided on the heatmap by different colors.

9) Lines 128-129: This sentence is difficult to understand. Does this mean that the gene expression in α cell-enriched pseudoislets returned to normal after 10 days of culture? Or genes differentially expressed in α cells at 5 and 10 days, were "unchanged" in pseudoislets made without cell sorting as well as whole islets?

We agree this sentence is complicated and is now edited to clarify our point. We meant by saying “expression levels of the altered genes returned to normal levels on day 10” that the genes differentially expressed in unsorted pseudoislets on day 5 vs day 0 were unchanged on day 10 vs day 0. We interpreted the data as expression levels of α- or β-enriched genes were recovered after typical islet architecture was regained in unsorted pseudoislets.

10) Figure 5b-d: Are these paired experiments comparing α cell-enriched pseudoislets with pseudoislets made of unsorted islet cells and native islets? This is very difficult to follow because the proportion of cell types has changed after α cell purification.

No, the data in Figure 5b-d do not compare α-pseudoislets with unsorted pseudoislets or with native islets since the proportion of cell types are not the same after sorting.

Figure 5b shows altered pathways both in α-pseudoislets and in unsorted pseudoislets with time.

Figure 5c shows altered pathways exclusively in α-pseudoislets with time.

Figure 5d shows altered genes exclusively in α-pseudoislets with time.

11) Figure 4d: Does this imply dispersion doesn't influence gene expression?

Yes, Figure 4d indicates that dispersion of islet cells into single cells did not cause acute changes in the gene expression other than expression levels of few genes (*DUSP5, CCN2, NR4A1*, and *NR4A2*). Dispersion-related changes in the gene expression levels were seen on day 5 and day 10 post-sorting.

12) Figures 4-5: The authors utilize the terms "day 0", "day 5", and "day 10", although the meaning of these terms is not unclear. Is this in reference to days post-sort or the number of days in culture after the pseudoislets have formed? If the former, clarity overall would be improved by referring to cells as "sorted α cells" and "unsorted islet cells" prior to pseudoislet formation. After reaggregation, referral to these as "α -pseudoislets", "unsorted pseudoislets", and "native islets" would be more appropriate. For whole islets, does "day 0" imply day of arrival?

Human islets were cultured overnight upon arrival and used for FACS the next day. The day of sorting is considered as day 0. So, days 0, 5, 10 indicate the number of days post-sorting. For whole islets, day 0 imply the next day of arrival. Methods section has been edited to clarify meaning of day 0, 5, 10.

Both the sorted α-cells and the unsorted islet cells formed islet-like clusters in a few days after sorting when plated in round-bottom not-treated plates. So, we edited the manuscript and referred to sorted cells as "sorted α cells" or "unsorted islet cells" prior to aggregation (on day 0). After reaggregation (on day 5 and day 10), we referred to sorted cells as "α -pseudoislets" or "unsorted pseudoislets". Instead of “whole islets”, we used "native islets" per suggestion.

13) Figure 2b: Is the time scale here (beginning with Day 1) have the same meaning as the time points in Figures 4 and 5?

Yes, the time scale used in Figure 2b is the same as in Figure 4 and Figure 5.

14) Figures 4a,5d: The color choice used to signify each donor makes the figure difficult to follow. More contrast between each assigned color would make it clearer.

Assigned colors have been changed in Figure 4a and 5d.

15) Lines 522-523: What is the magnification/scale bar size for the images in the top row of 2e?

Scale bar has been added to the top images. It is 100 μm.

16) Figure 5d: DLK1, GSN, and SMIN24 do not appear in the heatmap, though emphasized in the text.

Time-dependent changes in the levels of β-cell enriched genes, DLK1, GSN, and α-cell enriched gene SMIM24 were given in the heatmap in Figure 4a. We referred to Figure 4a in the results when emphasizing changes in DLK1, GSN, and SMIM24 in the updated manuscript.

17) Figure 5e: Order of insets for GCG and SMIM24 staining at Day 0 are switched compared to other panels.

Order of the insets has been corrected.

18) Lines 163-167: Are genes confirmed to also be downregulated at day 5, or was comparison only made between day 0 and day 10?

Yes, genes confirmed to be downregulated at day 5 as well. Comparison was made between day 5 and day 0, day 10 and day 0, day 10 and day 5. Consistently up or downregulated genes from day 0 to day 5, day 5 to day 10 were analyzed.

19) Lines 581-584: The stated description of data included in the heatmap is difficult to understand.

The statement has been edited.

20) Supplementary Figure 1c: Please provide the number of live cell pre-FACS as well as the number of live α cells recovered.

Approximately 15,000 IEQ were dissociated into single cells and sorted immediately. Live cell numbers were determined by trypan blue staining. The numbers in the table adjusted according to trypan blue staining results and live cells numbers were given for pre-FACS cells and for each sorted cell populations.

21) Supplementary Files 2,3, and 6 – More detailed data (reads, log2FC, FDR) for genes reported is required; listing "up" or "down" is unclear.

Supplementary Files 2,3,6 have been updated to provide log2FC and FDR values for the genes reported as up, down, or unchanged.

22) Methods: Was viability PI/DAPI stain used during the sort to gate on live cells (Supplementary Figure 1a)?

Viability stain was not used during the sorting of human islets. Live cell numbers were determined by trypan blue staining post-sorting.

23) Methods/Materials: Include catalog number for U-bottom plates used for pseudoislets.

Catalog number for the U-bottom plates (Corning #3788) has been added to Methods.

24) Methods: Are islet preps handpicked before sorting?

The human islets are not handpicked before sorting since the purity of the islet batches used in this manuscript is above 90%.

25) Methods: How many cells were counted to measure composition of α pseudoislets, unsorted pseudoislets, and whole islets (Figure 2f)?

The data given in Figure 2f in the first version of the manuscript represents an average of 4106. 4987 cells for α d5, 2919 cells for α d10, 3796 cells for unsorted d5, 4008 cells for unsorted d10, 6358 cells for whole islets d5, and 2568 cells for whole islets d10 were analyzed. Now, cells from additional 3 donors were analyzed in the current version and again ~4,000 cells were quantified per donor. The details of quantification of cell composition have been provided in methods under “Islet Immunohistochemistry and Quantification” section.

26) Methods: Are "unsorted" cells still labeled with DA-ZP1? If not, please discuss potential caveats of comparing unlabeled, unsorted cells with labeled cells.

Yes, unsorted cells are labeled with DA-ZP1 and kept on ice while the other cells were sorted by FACS. We have clarified this point in the methods section.

27) Methods: Please provide further detail on how expression level in 5f is calculated from images in 5e.

Details of quantification of expression levels have been provided in methods under “Islet Immunohistochemistry and Quantification” section.

28) Discussion: Please include a discussion of potential limitations to using this probe over antibody-based sorting approaches.

We have discussed both the advantages and potential limitations of using DA-ZP1 over antibody-based sorting approaches in the discussion.

29) References 21 and 22 are duplicated.

Duplicated references were removed.